# Characteristics and Classification of Choroidal Caverns in Patients with Various Retinal and Chorioretinal Diseases

**DOI:** 10.3390/jcm11236994

**Published:** 2022-11-26

**Authors:** Xiaohong Guo, Yao Zhou, Chenyang Gu, Yingjie Wu, Hui Liu, Qing Chang, Bo Lei, Min Wang

**Affiliations:** 1Henan Eye Hospital, Henan Eye Institute, People’s Hospital of Zhengzhou University, Henan Provincial People’s Hospital, Zhengzhou 450003, China; 2Department of Ophthalmology and Visual Science, Eye, Ear, Nose and Throat Hospital, Shanghai Medical College of Fudan University, Shanghai 200031, China

**Keywords:** swept-source optical coherence tomography, pachychoroid spectrum diseases, choroidal cavern, central serous chorioretinopathy, age-related macular degeneration

## Abstract

Purpose: To investigate the features of choroidal caverns in diverse retinal diseases with swept-source optical coherence tomography (SS-OCT). Methods: Subjects with normal eyes, retinitis pigmentosa (RP), wet age-related macular degeneration (wAMD), acute central serous chorioretinopathy (CSC), or chronic CSC were enrolled. The characteristics of choroidal caverns were evaluated with SS-OCT. The prevalence of choroidal caverns in retinal diseases and the correlations between the number, width and depth of choroidal caverns with the thickness of choroid were analyzed. Results: Among 315 eyes of 220 subjects, choroidal caverns were found in 110 eyes (34.9%). Choroidal caverns were divided into two categories based on their location and size. Type I was small and usually lobulated, presented in the choroidal capillary and Sattler’s layers. Type II was larger, usually isolated, and presented in the Sattler’s and Haller’s layers. The prevalence of type I in subjects with normal eyes, RP, wAMD, acute CSC, or chronic CSC was 17.4%, 19.6%, 1.6%, 32.8%, and 85.2%, respectively, while that of type II was 0%, 0%, 21.3%, 13.8%, and 53.7%, respectively. The number, width, and thickness of type II choroidal caverns correlated positively with macular choroidal thickness. Conclusions: Choroidal caverns could be divided into two categories. Type II choroidal caverns appeared associated with the pachychoroid spectrum and RPE atrophic diseases.

## 1. Introduction

The development of the optical coherence tomography (OCT) imaging technology, especially the enhanced depth imaging [1,2] and swept-source OCT (SS-OCT) [3,4,5], has made it possible to detect the fine structure of choroid. Choroidal caverns, small lesions within the choroid, were first discovered in 2016 by Querques et al. [6] in patients with geographic atrophy (GA). On OCT, choroidal caverns are located mainly in the Sattler’s and Haller’s layers of the choroid, appearing as circular areas of low reflectivity with regions of high reflectivity trailing behind them. Several studies have shown a dotted or linear area of high reflectivity in the caverns [1], whereas others denied the presence of such areas [7]. Although choroidal caverns usually occur in patients with GA [8,9,10], pachychoroid spectrum disease [11,12,13,14,15], retinal pigment epithelium (RPE) atrophy diseases such as Stargardt disease [16], best vitelliform dystrophy [17] and rod–cone dystrophy [7,18], they are also seen in normal eyes or choroidal osteoma [19]. Although previous studies have described the shape and size of choroidal caverns, they have not systematically analyzed the related characteristics of choroidal caverns. We observed the morphology, size, and distribution of choroidal caverns in several chorioretinal diseases.

## 2. Methods

This was a retrospective observational study. Patients who visited the Eye and Ear, Nose, and Throat Hospital of Fudan University, Shanghai and Henan Eye Hospital, Zhengzhou between January 2019 and November 2021 were enrolled and divided into seven groups based on their ocular condition. The grouping and inclusion criteria were the following: (1) normal eyes, without eye disease and equivalent spherical degree > −6D; (2) retinitis pigmentosa (RP); (3) wet age-related macular degeneration (wAMD) with a course of >6 months; (4) acute central serous chorioretinopathy (CSC) with focal or diffuse choroidal thickening on OCT and dilated choroidal vessels on en face SS-OCT, with a course of ≤3 months, and (5) chronic CSC with similar OCT findings to acute CSC, with a course of ≥6 months. The exclusion criteria were the following: (1) disease complicated with other congenital or developmental ocular abnormalities, such as typical retinochoroidal coloboma, morning glory syndrome, retinal or choroidal tumor, uveitis, scleritis, and other ocular diseases; (2) history of ocular trauma; (3) medium opacity; (4) diabetic retinopathy; (5) history of hypertension. The research design adhered to the principles of the Declaration of Helsinki and was approved by the Ethics Committees of the Eye and ENT Hospital of Fudan University (protocol code: 2019099) and Henan Eye Hospital (protocol code: HNEECKY-2022(38)).

All eyes were examined with SS-OCT and color fundus photography (CFP). Some patients with CSC and wAMD were examined with fundus fluorescein angiography (FFA) combined with indocyanine green angiography (ICGA). The VG 200D SS-OCT system (Svision Imaging, Luoyang, Henan, China) was used for image acquisition, including a 12 mm × 12 mm volume scan with a high-definition B-scan and en face OCT [20]. “Choroidal cavern” was defined as a region of low reflectivity within the choroid, with or without a trailing region of high reflectivity [1,2]. In the enhancement mode of the system, image contrast was enhanced [21], making it easier to observe the low-reflective choroidal caverns and to distinguish them from other low reflectivity lesions (Figure 1). The upper and lower boundaries of the choroidal caverns were defined as their distance to the Bruch’s membrane (Figure 2).

The morphology, location and size of choroidal caverns were observed. The choroidal thickness of the fovea and the horizontal diameter and depth of the largest cavern in each eye were measured with the caliper tool in VG 200D. Spectralis HRA (Heidelberg Engineering GmbH, Heidelberg, Germany) was used for FFA and ICGA. En face OCT images were superimposed on FFA and ICGA images to observe the features of the choroidal caverns. The presence or absence of caverns and their classification were determined by two observers (G.X. and Z.Y.) with same criteria.

SPSS 25.0 software was used for statistical analyses, including the occurrence of different types of choroidal caverns, the correlation between the size of the type II choroidal caverns and the choroidal thickness. Descriptive statistics are reported as frequency (percentage, %) and mean (standard deviation, SD). The prevalence differences related to two or more groups were tested with Chi-square test or Fisher’s exact test. The correlation between choroidal thickness and choroidal caverns was analyzed with a linear correlation analysis. *p* < 0.05 was considered statistically significant.

## 3. Results

### 3.1. Characteristics and the Clinical Classification of Choroidal Caverns

Among 315 eyes of 220 patients, choroidal caverns were found in 110 eyes, with a prevalence of 34.9%. Based on their location and distribution, we divided the choroidal caverns into two categories.

(1)Type I were small and usually lobulated, occurred in the choroidal capillary and the Sattler’s layers. Depending on the overall choroidal thickness, the upper margin was around 25~50 μm below the Bruch’s membrane (BM) and the lower margin was around 150~200 μm below BM. Furthermore, choroidal caverns outside the macular area were designated subtype Ia (Figure 2) and those within the macular area were designated subtype Ib (Figure 3).

(2)Type II were large, usually isolated and in the Sattler’s and Haller’s layers. Depending on the overall choroidal thickness of the patients, the upper margin was around 100~200 μm below the BM and the lower margin was around 200~400 μm below the BM (Figure 4).

### 3.2. Characteristics of Type I Choroidal Caverns

Type I choroidal caverns were detected in 92 eyes, with a total prevalence of 29.2%, including 35 eyes with type Ia, 28 eyes with type Ib, and 29 eyes with both type Ia and Ib caverns. The prevalence of eyes with type I caverns in normal, RP, wAMD, acute CSC, or chronic CSC eyes was 17.4%, 19.6%, 1.6%, 32.8%, or 85.2%, respectively. A χ^2^ test (crossover analysis) showed a significant difference in the prevalence of type I choroidal caverns between these patient groups (χ^2^ = 76.848, *p* = 0.000). Among these patients, the prevalence of type I choroidal caverns was highest in those with chronic CSC, followed by acute CSC.

Type Ia choroidal caverns were predominantly outside the macular area, near the upper and lower vascular arches. Choroidal caverns were visible around the large choroidal vessels, with irregular shapes and clustered distributions (Figure 2). Type Ia choroidal caverns were predominant in subjects with normal eyes or acute CSC. Type Ib caverns in the macular region (Figure 3) occurred in 54.6% (6/11) of RP patients (Figure 5). Both type Ia and Ib were the most common in patients with chronic CSC (Table 1 for the classification of type I caverns in each group of patients). Type I choroidal caverns had no blood flow signal on OCTA tomography. They were detected in the choroidal capillary layer and the Sattler’s layer. There was no punctate or linear internal hyperreflectivity. No obvious abnormalities (Figure 6) were detected with fundus color photography, FFA, or ICGA.

### 3.3. Characteristics of Type II Choroidal Caverns

Type II choroidal caverns were detected in 50 eyes, with a total prevalence of 15.9%. No type II choroidal caverns were found in the normal or RP eyes. The prevalence of type II caverns in wAMD, acute CSC, or chronic CSC patients was 21.3% (13/61), 13.8% (8/58), and 53.7% (29/54), respectively. A χ^2^ test (cross analysis) showed significant difference in the prevalence of type II choroidal caverns among these patient groups (χ^2^ = 71.750, *p* = 0.000). The mean diameter of the type II choroidal caverns was 394.5 ± 287.1 μm, the mean depth was 216.1 ± 118.5 μm, and the mean number per eye was 4.2 ± 4.9 (Table 1).

Furthermore, the width, depth, and number of type II choroidal caverns differed significantly among the three disease groups. The width and depth of the type II choroidal caverns decreased in the following order: chronic CSC > acute CSC > wAMD. The number of type II choroidal caverns decreased in the following order: chronic CSC > wAMD > acute CSC. Type II choroidal caverns were located in the Sattler’s and Haller’s layers (Figure 3 and Figure 7) of the choroid, and they were not overlapped with blood flow signal on OCTA. There was no punctate or linear hyperreflectivity in the caverns of patients with wAMD. However, punctate or linear hyperreflectivity was detected in 40.5% (15/37) of acute or chronic CSC patients with the type II choroidal caverns. There was no hyperfluorescence or hypofluorescence (Figure 4) corresponding to choroidal caverns on CFP, FFA, or ICGA. The width, thickness, and number of all type II choroidal caverns correlated positively with the subfoveal choroidal thickness (Figure 8).

## 4. Discussion

Since being first documented by Querques and colleagues in GA eyes, choroidal caverns have been increasingly recognized in several chorioretinal diseases, largely because of the application of the advanced high penetrating OCT instruments which make detection of lesions in the choroid possible [6,7]. The prevalence of choroidal caverns differs across diseases, ranging from 12.5% to 43.9% [6,7]. In this study, the overall prevalence of choroidal caverns in a cohort of subjects was 34.9%. Nevertheless, the prevalence of choroidal caverns in the diseased eyes was higher than that of the normal eye group, suggesting choroidal caverns may represent pathologic biomarkers in the choroid.

The number, size and shape of choroidal caverns were significantly different among each other. It appeared that these features were largely associated with the location of choroidal caverns and the depth of choroid. To describe the image characteristics better, we proposed to classify choroidal caverns into two categories, type I and type II. Type I were small and usually lobulated, occurred in the choroidal capillaries and the Sattler’ layer. Type II were large, usually isolated and in the Sattler’s and Haller’s layers.

The prevalence of type I choroidal caverns in patients with normal eyes, RP, wAMD, acute CSC, or chronic CSC was 17.4%, 19.6%, 1.6%, 32.8%, or 85.2%, respectively. Obviously, the prevalence of type I choroidal caverns in patients with chronic CSC was significantly higher. When the distribution of type I choroidal caverns was considered, wAMD showed only type Ib caverns, whereas all other patient groups showed both type Ia and Ib caverns.

Type II choroidal caverns were only found in patients with wAMD, acute CSC, or chronic CSC, with prevalence rates of 21.3%, 13.8%, or 53.7%, respectively. The width, depth, and number of choroidal caverns correlated significantly with the thickness of the choroid. In the previous studies in which no classification was applied, Dolz-Marco et al. [7] showed a mean cavern size of 61 μm and Friedman and Smith [22] reported that the size ranged from 25 μm to 150 μm. In Sakurada’s study, eyes presented multiple caverns ranging from 249 to 486 μm [11]. Nevertheless, we did not measure the size and number of type I choroidal caverns, since they were small and usually lobulated. However, each type II choroidal cavern exhibited independently and was significantly larger in size. The maximum width diameter of type II choroidal caverns was from 70 μm to 1588 μm, and the depth was from 39 μm to 537 μm, which was larger than the maximum choroidal caverns measured in previous studies. The difference between this and previous studies might be a consequence of the fact that high penetrating instrument was applied in the current study.

There are two main theories about the mechanisms of choroidal caverns. In the vascular degeneration theory [11], the caverns arise from the non-perfused ghost blood vessels and the persistence of the matrix column in which the blood vessels were originally located. Querques [23] proposed that in myopic patients, the lateral, anterior, and posterior stretching of the choroidal tissue and a reduction in the density of choroidal blood vessels led to the formation of choroidal caverns, so these caverns might represent regions lacking blood vessels and matrix. The other theory of abnormal lipid metabolism in the outer retina [7] suggests that although the metabolism of photoreceptor cells consumes lipid substances, the proteins and substances required for lipid metabolism are not expressed in the outer retina but are strongly expressed in the choroid [24,25]. The choroid stores lipid droplets, which may ensure the normal metabolism of the outer retina. In fact, in 1934, Jaensch demonstrated the presence of fat deposits in the choroidal matrix of eyes of different ages and with different disorders. Animals and humans have a complex, finely regulated, long-term lipid nutrition delivery system [24], which is in equilibrium in healthy condition.

Our observations tend to decline the vascular degeneration theory. If the formation of choroidal caverns was caused by the degeneration and occlusion of blood vessels, then obliterative fibrous intimal thickening with collagen hyperplasia should be detected histologically, resulting in a mild high reflection boundary on OCT images. Nevertheless, no reflective signal was observed [7]. In addition, observation of adjacent choroidal caverns with OCT and en face OCT (Figure 4) suggested that the punctate or linear internal hyperreflectivity was the space formed by the normal choroidal matrix between adjacent caverns.

On the other hand, our data were in favor of the hypothesis of abnormal lipid metabolism. Indeed, a pathological study also supported this notion [7]. Yoichi et al. [26] treated a CSC patient with subcentral choroidal caverns with half-dose verteporfin-based photodynamic therapy. After 3 months, the neuroepithelial effusion disappeared, the choroidal thickness decreased significantly, and the choroidal caverns disappeared. A possible interpretation was that the hyper perfusion state of choroid and the PRE function improved and the lipid caverns that had accumulated in the choroid disappeared.

The prevalence of choroidal caverns, whether type I or type II, was significantly higher in patients with CSC, especially those with chronic CSC, than in those with other diseases. Patchy choroidal disease is characterized by a focal or diffuse increase in the choroidal thickness, which is caused by Haller choroidal vasodilation. CSC is a common disease in the spectrum of patchy choroidal diseases [27]. Swollen and permeable choroidal vessels and damaged RPE are considered to be important factors in promoting the occurrence and development of CSC [28]. We speculate that with an increase in choroidal thickness and destruction of the RPE, the lipid transport function of RPE was damaged and large amount of lipid could be accumulated in the choroidal matrix. In the study of Dolz-Marco [7], the choroidal lipid droplets characteristically showed either hyperfluorescence or hypofluorescence on ICGA. In our study, FFA and ICGA images of some CSC patients were compared with their OCT images, but no corresponding angiographic manifestations with choroidal caverns were observed. The fluorescence in the areas corresponding to choroidal lipid droplets still depended mainly on the state of the RPE. Nevertheless, the lesion of choroidal caverns may not be consistent with the area with previous retinal serous detachment. For example, the choroidal caverns could be outside of the area with previous retinal serous detachment [29].

In our study, the prevalence of type II choroidal caverns in patients with wAMD was 21.3%, which was lower than in chronic CSC patients but higher than in acute CSC patients. The patients with wAMD had been diagnosed more than 6 months earlier. The function of RPE cells was clearly damaged and might have affected the transportation of lipid. Although the prevalence of type II choroidal caverns was higher in patients with wAMD than in those with acute CSC, the width, depth, and number of type II choroidal caverns were smaller than in patients with acute CSC.

Currently, there is still no evidence that there are connections between type I and type II choroidal caverns. Apparently, type I could be seen in both normal and diseased eyes, while type II was evident in the diseased eyes. Although several studies have suggested that the etiology of choroidal caverns might be associated with abnormal lipid metabolism, convictive evidence remains limited. Based on our research and literature review, we propose three possible explanations for the formation and distribution of choroidal caverns, as shown in Figure 9. (1) In normal eyes with no retinal or choroidal disease, choroidal caverns present with a low occurrence. They are distributed mainly near the vascular arch of the posterior pole or in the macular area and are mainly type I. (2) In patchy choroidal spectral diseases, the perfusion of the choroid increases the prevalence of type I choroidal caverns, and large, single, type II choroidal caverns occur. (3) Atrophic changes of RPE, such as in wAMD, are highly associated with type II choroidal cavern. Therefore, choroidal cavities are common in RPE atrophy diseases and in patchy choroidal diseases.

Our study had several limitations. First, although we included five common diseases associated with choroidal caverns, a larger sample size with more diverse diseases is warranted for a detailed prevalence of choroidal caverns in the diseased population. In this pilot research, we did not detect type II choroidal caverns in normal and RP eyes. A larger sample size would be helpful to elaborate on the prevalence of choroidal caverns in these conditions. Second, a follow-up observation of choroidal caverns is desired. Finally, since the main aim of the current study was to explore the association between the diseases and choroidal caverns, we did not discuss the relation between the lesion and age, as well as other factors.

In conclusion, we studied the types and distribution of choroidal caverns, as well as their correlation with specific retinal diseases. Although choroidal caverns could be observed in normal eyes, the significantly higher prevalence suggested that they were associated with the hyper perfusion of the choroid and atrophy of the RPE. Their clinical significance deserves further investigation.

## Figures and Tables

**Figure 1 jcm-11-06994-f001:**
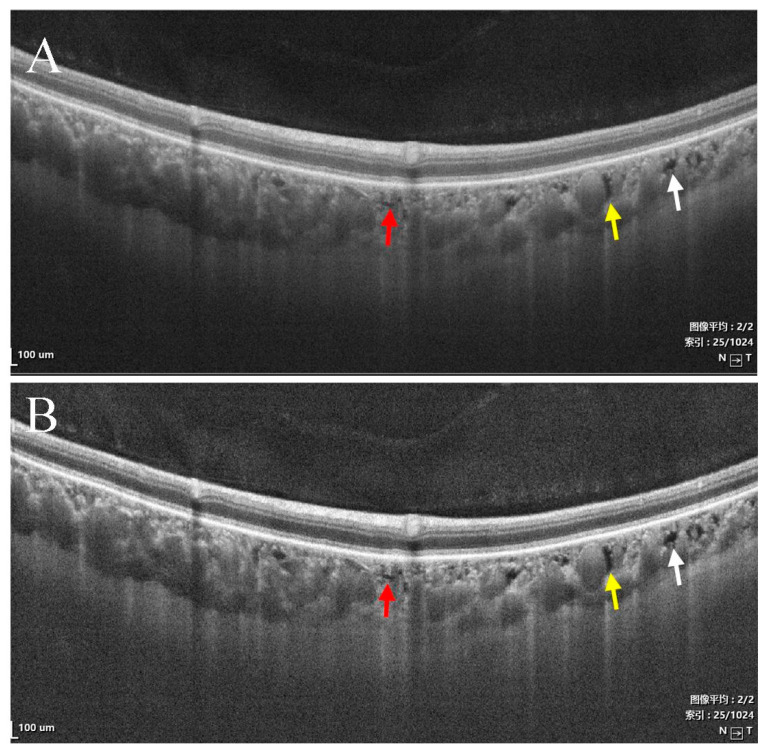
SS-OCT showed choroidal vessels and choroidal caverns without (**A**) and with enhanced mode (**B**). Large choroidal vessels have oval lumens with highly reflective walls (yellow arrows). Choroidal caverns appear as irregular low reflection signals within the choroidal vessel layers. After enhancement, the choroidal caverns were visible (red and white arrows).

**Figure 2 jcm-11-06994-f002:**
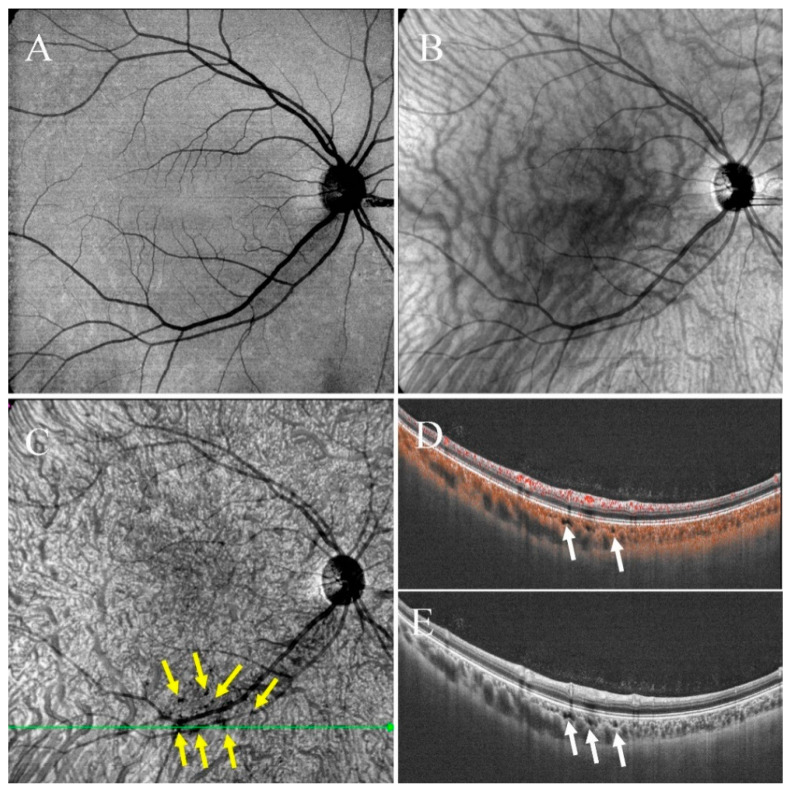
Multimodal images of right eye of a 34-year-old woman diagnosed with normal eye. From the en face structural OCT of choroidal capillary layer (**A**) and choroid layer (**B**), we could not detect any evidence of choroidal caverns. (**C**) En face structural OCT (12 mm × 12 mm; upper boundary was about 50 μm below BM and lower boundary was about 140 μm below BM) showing clusters of choroidal caverns around the large choroidal vessels (yellow arrows). (**D**,**E**) Choroidal caverns showed no signal on OCTA. Hyporeflective structures located within the choroidal capillary and the Sattler’s layers (white arrows).

**Figure 3 jcm-11-06994-f003:**
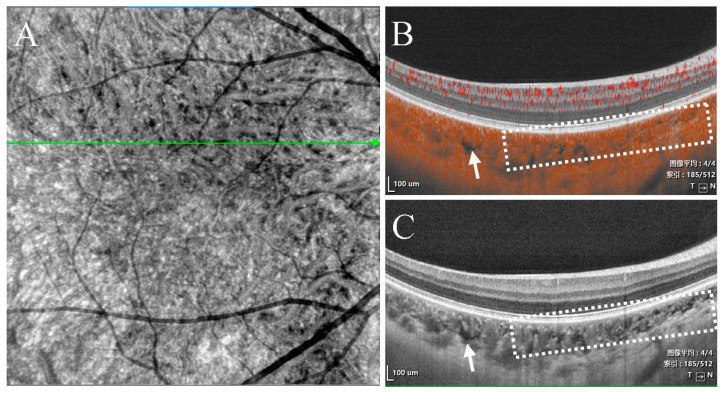
Multimodal images of the right eye of a 24-year-old woman with normal vision showing type Ib choroidal caverns. (**A**) En face structural OCT (6 mm × 6 mm; upper boundary was about 25 μm below BM and lower boundary was about 150 μm below BM) showing clusters of choroidal caverns around the large choroidal vessels. (**B**,**C**) Choroidal caverns showed no signal on OCTA. Hyporeflective structures located within all layers of the choroidal vessels (white arrows and white dashed box).

**Figure 4 jcm-11-06994-f004:**
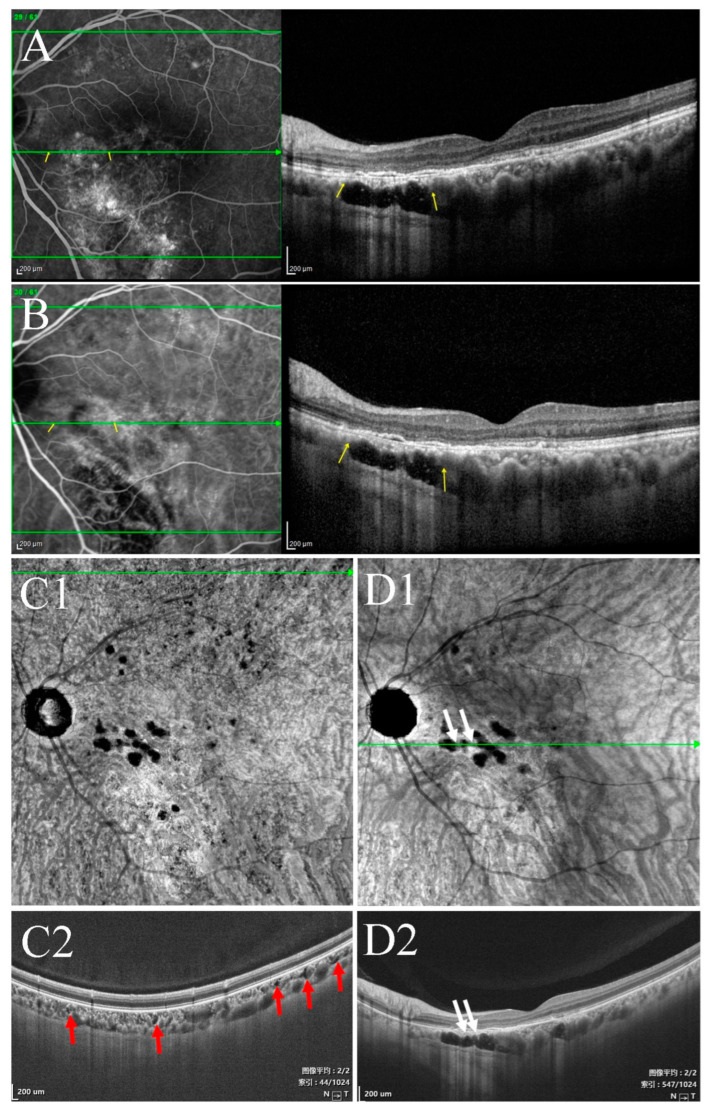
Multimodal images of the left eye of a 51-year-old man diagnosed with chronic CSC 3 years ago. (**A**,**B**) Structural OCT shows type II choroidal caverns. The boundary of the lesion was marked by the yellow arrows. In the macular area, FFA (**A**) and ICGA (**B**) showed hyperfluorescence or hypofluorescence that was inconsistent with the scope of the lesion. (**C1**,**C2**) En face OCT (upper boundary was around 50 μm below BM and lower boundary was around 150 μm below BM) and structural OCT showed type Ia choroidal caverns (red arrows). (**D1**,**D2**) Type II choroidal caverns appeared as areas of hyporeflection on en face OCT (the upper boundary was around 50 μm below BM and the lower boundary was 300 about μm below BM), with fuzzy boundaries between choroidal caverns and a hyperreflective column between two adjacent choroidal caverns.

**Figure 5 jcm-11-06994-f005:**
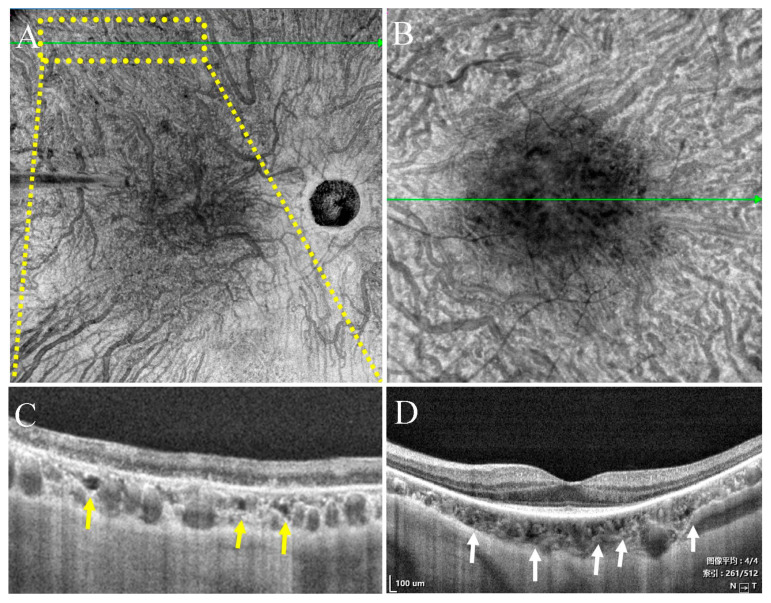
OCT of a patient with retinitis pigmentosa. (**A**,**C**) Type Ia choroidal caverns (yellow arrow). (**B**,**D**) Type Ib choroidal caverns (white arrows).

**Figure 6 jcm-11-06994-f006:**
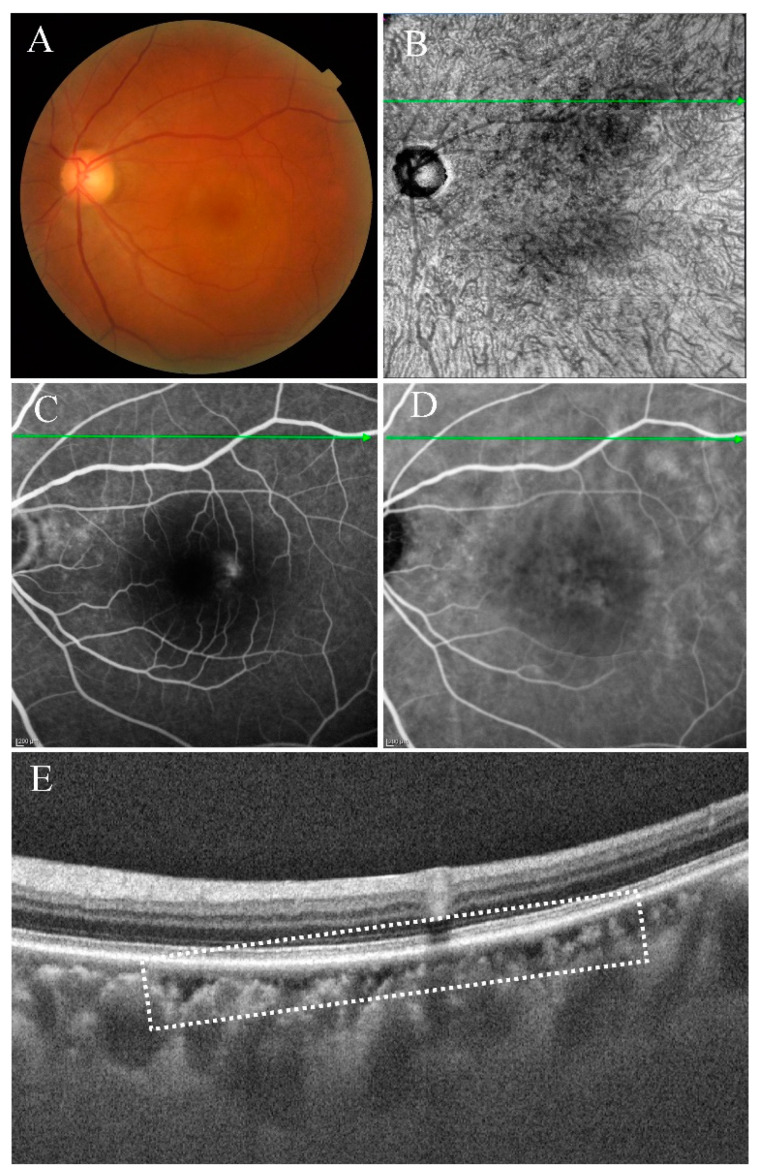
Multimodal images of the left eye of a 58-year-old man diagnosed with acute CSC 2 months ago. (**B**) En face structural OCT (upper boundary was about 50 μm below BM and lower boundary was around 100 μm below BM) and structural OCT (**E**) showed type Ia choroidal caverns (white dashed box). There was no abnormality on CFP (**A**), FFA (**C**), or ICGA (**D**).

**Figure 7 jcm-11-06994-f007:**
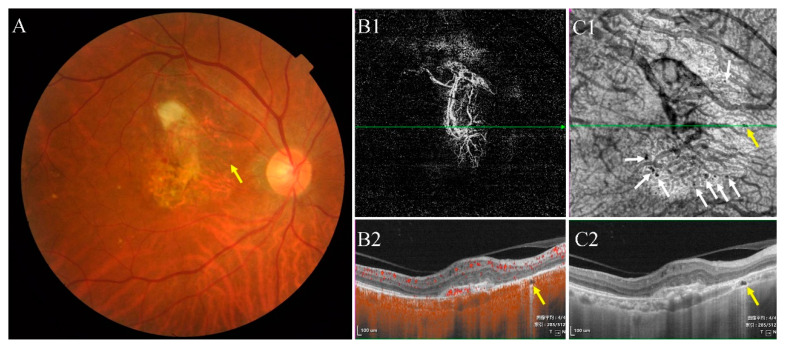
Multimodal images of the right eye of a 73-year-old man diagnosed with wAMD 2 years ago. (**A**) Color fundus photograph showed yellow and white lesions. (**B1**,**B2**) The morphology of neovascularization in the outer retina layer. (**C1**) En face structural OCT (the upper boundary was about 30 μm below BM and the lower boundary was about 105 μm below BM) showed type II choroidal caverns (white and yellow arrow). (**C2**) Structural OCT scan of the corresponding areas in (**B1**,**C1**) (green line) showed choroidal caverns among the Sattler’s and Haller’s layers (yellow arrows).

**Figure 8 jcm-11-06994-f008:**
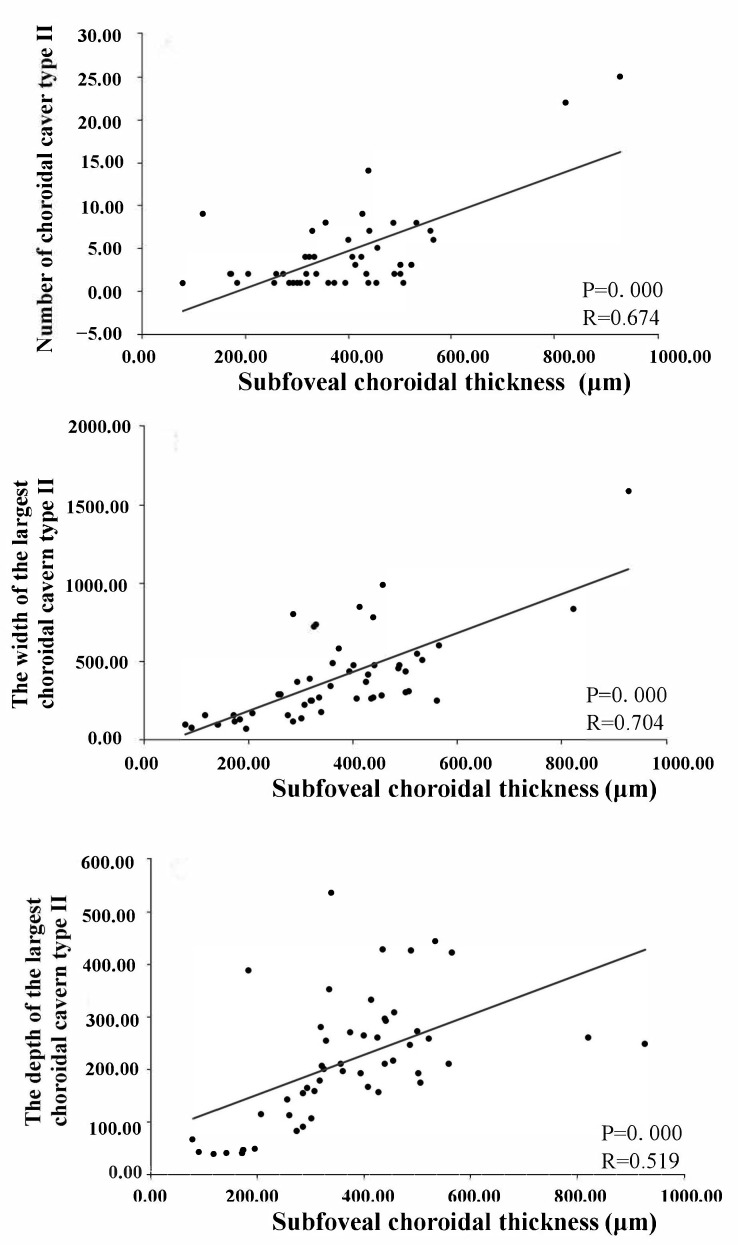
Relationship between the number of type II choroidal caverns, the width of the largest type II choroidal cavern, the depth of the largest type II choroidal cavern, and the subfoveal choroidal thickness.

**Figure 9 jcm-11-06994-f009:**
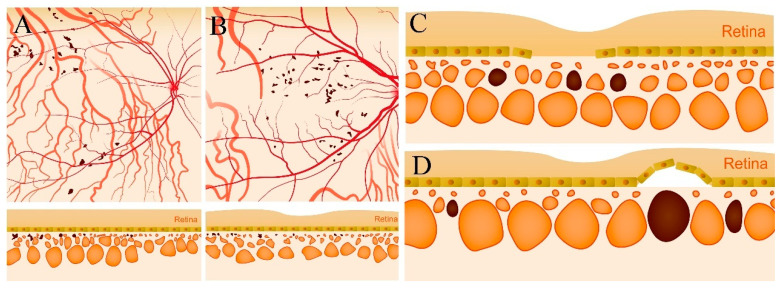
Diagram of the distribution and characteristics of choroidal caverns. (**A**) Type Ia choroidal caverns outside the macular area are small, mostly lobulated, and located in the choroidal capillaries and the Sattler’s layer. (**B**) Type Ib choroidal caverns in the macular area are small, mostly lobulated, and located in the choroidal capillaries and the Sattler’s layer. (**C**) Type II choroidal caverns in patients with RPE atrophy are much larger, isolated, and in the Sattler’s and Haller’s layers. (**D**) Type II choroidal caverns in patients with pachychoroid spectrum disease are the largest, isolated, and in the Sattler’s and Haller’s layers (orange represents choroidal blood vessels and dark brown represents choroidal caverns).

**Table 1 jcm-11-06994-t001:** Demographic and clinical characteristics of the subjects with the choroidal caverns.

	Normal	RP	wAMD	Acute CSC	Chronic CSC	Total
Eyes/People	86/49	56/29	61/51	58/58	54/33	315/220
Male/Female	20/29	14/15	37/14	41/17	30/3	142/78
Age, yrs, mean ± SD	44.3 ± 13.9	44.4 ± 20.0	69.1 ± 8.6	45.5 ± 8.9	50.3 ± 8.3	51.3 ± 15.5
Refractive error of eyes, diopter, median (range)	−2.5 (−5.0–0)	−1.0 (−5.25–1.75)	−0.5 (−4.75–2.5)	−1.25 (−3.75–1.0)	−1.0 (−4.0–2.25)	-
BCVA, median (range)	20/20 (20/25–20/20)	20/100 (20/200–20/66)	20/100 (20/200–20/50)	20/50 (20/100–20/25)	20/100 (20/200–20/50)	-
Presence with choroidal cavern type I, n (%)	15 (17.4%)	11 (19.6%)	1 (1.6%)	19 (32.8%)	46 (85.2%)	92 (29.2%)
choroidal cavern type Ia	9	5	0	12	9	35
choroidal cavern type Ib	6	6	1	4	11	28
choroidal cavern ype Ia + Ib	0	0	0	3	26	29
Presence with choroidal cavern type II, n (%)	0	0	13 (21.3%)	8 (13.8%)	29 (53.7%)	50 (15.9%)
The width of the largest choroidal cavern type II, (μm, mean ± SD)	0	0	197.6 ± 197.7	282.0 ± 138.0	513.7 ± 294.0	394.5 ± 287.1
The depth of the largest choroidal cavern type II, (μm, mean ± SD)	0	0	83.7 ± 48.6	252.1 ± 134.0	265.6 ± 87.3	216.1 ± 118.5
Number of choroidal cavern type II, (n, mean ± SD)	0	0	2.0 ± 2.2	1.5 ± 1.0	5.9 ± 5.8	4.2 ± 4.9
Choroidal thickness (μm, mean ± SD)	259.8 ± 80.2	211.9 ± 61.3	200.9 ± 80.4	365.4 ± 90.7	440.9 ± 134.5	290.3 ± 126.4

Yrs, years; n, number; BCVA, best corrected visual actuality; Post-op RD, post operation retinal detachment; RP, retinitis pigmentosa; wAMD, age-related macular degeneration; CSC, central serous chorioretinopathy.

## Data Availability

The data presented in this study are available on request from the corresponding author. The data are not publicly available due to privacy and ethical reasons.

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
