# Peer review of "Characteristics and Classification of Choroidal Caverns in Patients with Various Retinal and Chorioretinal Diseases"

_jcm, 2022, doi:10.3390/jcm11236994_

Round 1
Reviewer 1 Report (Previous Reviewer 1)
I would like to thank the authors for preparing the revised version which is noticeably improved.
Author Response
Thank you for your time and efforts for reviewing this manuscript.
Reviewer 2 Report (New Reviewer)
The authors presented an interesting retrospective study entitled “Characteristics and Classification of Choroidal Caverns in Patients with Various Retinal and Chorioretinal Diseases”. It was based on an evaluation of 315 eyes of 220 patients to investigate the choroidal caverns, which belong to the entity of the choroidal spectrum lesions. The paper is of potential clinical interest. I recommend just minor changes:
- The Reference section is not sufficient for this type of paper.
- The authors should explain whether the patients' previous treatment (anty-VEGF therapy) could have an effect on these lesions?- Detailed statistical methodology should be described.
Author Response
1- The Reference section is not sufficient for this type of paper.
RESPONSE: Thanks for your critiques. We have added references relevant to the text.
2- The authors should explain whether the patients' previous treatment (anty-VEGF therapy) could have an effect on these lesions?
RESPONSE: Thanks for your careful reading and insightful comments.
It is true that some patients were treated with anti-VEGF before coming to our hospital, but we couldn’t get their previous OCT imaging data, which were in the other hospitals. In the cases we observed, choroidal caverns mignt not be affected by anti-VEGF. We agree it is an interesting question and deserve further studies.
3- Detailed statistical methodology should be described
RESPONSE: Thanks for your suggestions. We have revised the statistical methodology carefully (page 3).
This manuscript is a resubmission of an earlier submission. The following is a list of the peer review reports and author responses from that submission.
Round 1
Reviewer 1 Report
I would like to thank the authors for their effort, however, several issues need to be addressed:
1. Last paragraph of the introduction section:"we noticed that the shape and size of choroidal caverns in the choroid were remarkably different. We speculated that the characteristics and clinical significance of these choroidal caverns were different. Therefore, we proposed to classify choroidal caverns base on the location and analyzed their features including incidence, distribution, size, in various chorioretinal diseases." The authors seem to present a part of their results in the introduction section. The introduction section should not present the results of current study.
2. The literature review and given evidence in the introduction and discussion sections is inadequate. Looking at the total number of references (12 references including self citations) clearly reflects this.
3. The authors have included a wide variety of patients in this study(high myopia, post-op RD, RP, wAMD, acute CSC, or chronic CSC). The diversity in the diseases included makes it hard for the authors to focus and explain the main aim of the study throught the manuscript. Even the title of the article does not cover all of this entities although the authors have tried to include a wide range of diseases in the title.
4. The authors have only prepared one table to present all the demographic, clinical and OCT measurements of the study. The authors would better to rearange the table.
5. The incidence of type 2 is reported to be 0% in four groups. This might be due to the small number of included patients in the study. The authors would better to mention this in the limitations.
Author Response
Dear Editor and Reviewers,
We would thank the reviewers for their constructive, insightful comments and suggestions. We have carefully addressed their concerns and made relevant corrections in the text. We believe the manuscript is improved after taking their comments.
Thanks again.
Reviewer Comments:
Reviewer
- Last paragraph of the introduction section: "we noticed that the shape and size of choroidal caverns in the choroid were remarkably different. We speculated that the characteristics and clinical significance of these choroidal caverns were different. Therefore, we proposed to classify choroidal caverns base on the location and analyzed their features including incidence, distribution, size, in various chorioretinal diseases." The authors seem to present a part of their results in the introduction section. The introduction section should not present the results of current study.
RESPONSE: Thanks for your suggestion. We have made corresponding changes in the article. “Although previous studies have described the shape and size of choroidal caverns, they have not systematically analyzed the related characteristics of choroidal caverns. Therefore, we observed the morphology, size, and distribution of choroidal caverns in several choroidoretinal diseases.” (line 49-52)
- The literature review and given evidence in the introduction and discussion sections is inadequate. Looking at the total number of references (12 references including self citations) clearly reflects this.
RESPONSE: Thanks for your critiques. We have made changes and reorganized the references.
4 Xia YH, Feng NJ, Hua R. “Choroidal caverns” spectrum lesions. Eye. 2021;35(5)1508-1512. DOI: 10.1038/s41433-020-1074-y.
5 Ayachit A, Joshi S, Kathyayini SV, et al. Choroidal caverns in pachychoroid neovasculopathy. Indian J Ophthalmol. 2020.68(1):199-200. DOI: 10.4103/ijo.IJO_395_19.
6 Pederzolli M, Sacconi R, Battista M, et al. Bilateral choroidal caverns in a child with pachychoroid and anxious personality. Am J Ophthalmol Case Rep. 2022;26:101505. DOI: 10.1016/j.ajoc.2022.101505.
7 Dario PM, Dario G, Myrta L, et al. Choroidal Caverns in Stargardt Disease.Retina. 2022;63(2):25. DOI: 10.1167/iovs.63.2.25.
8 Carnevali A, Sacconi R, Corbelli E, et al. Choroidal caverns: a previously unreported optical coherence tomography finding in best vitelliform dystrophy. Ophthalmic Surg Lasers Imaging Retina. 2018;49:284–287. DOI: 10.3928/23258160-20180329-14
- Shoji Kishi, Hidetaka Matsumoto. A new insight into pachychoroid diseases: Remodeling of choroidal vasculature. Graefe's Archive for Clinical and Experimental Ophthalmology. 23 January 2022. Doi.org/10.1007/s00417-022-05687-6
- The authors have included a wide variety of patients in this study (high myopia, post-op RD, RP, wAMD, acute CSC, or chronic CSC). The diversity in the diseases included makes it hard for the authors to focus and explain the main aim of the study throught the manuscript. Even the title of the article does not cover all of this entities although the authors have tried to include a wide range of diseases in the title.
RESPONSE: Thank you for your comments. You are right. We have indeed struggled for a long time for making an adequate title for the manuscript. But the title would be too long had we added all entities to it. We would be grateful if the reviewer could help us on this.
- The authors have only prepared one table to present all the demographic, clinical and OCT measurements of the study. The authors would better to rearange the table.
RESPONSE: Thanks for your comments. Since we have many figures, we have put the necessary information in one table. We hope it would be helpful for the readers to compare the different characteristics between each group.
- The incidence of type 2 is reported to be 0% in four groups. This might be due to the small number of included patients in the study. The authors would better to mention this in the limitations.
RESPONSE: Thanks for your suggestion. We have added this to the limitations.
“In this pilot research, we didn’t find type 2 choroidal caverns in normal, high myopia, post-op RD and RP eyes. A larger sample size would be helpful to elaborate the incidence of choroidal caverns in these conditions .”(line 340-343)

Reviewer 2 Report
This study, although not original, is rather interesting as it addresses a relatively new and controversial finding in the choroid, includes a significant sample of patients and has good quality imaging.
Nevertheless, there are several important issues needing
1 - The manuscript would benefit from a proofread by an english-native speaker.
2 - Diabetes and hypertension were not exclusion criteria and should have been as they can cause microvascular choroidopathy
3 - The authors do not specify the exact methodology in segmenting the caverns. Who performed it? Was it performed by one or multiple observers?
3 - The authors do not specify the cause of retinal detachment and neither if the macula was on or off. They also mention that only type Ia lesions were found but do not specify if they were spatially related with the pre-operative detached retinal area or not. By the demographic data on table 1 we understand that some of those patients had high myopia as a probable cause of RD and that causes a bias in the sub-group analysis.
4 - Patients with high myopia have very thin choroids. The outer boundary of 110~150 to determine if it is I or II type lesion could not apply to these patients as Haller vessels are frequently located at this level in highly myopic patients.
5 - Furthermore, according to the authors definition (in page 3 - 5), patient 8 should have a type I and not type II lesion due to the location of the lesion (“upper boundary is 30 μm below BM and lower boundary is 105 μm below BM”) . To me this is the most worrisome issue in this article. The definition of the 2 types of lesion might be ambiguous.
6 - There was not a statistical adjustment of the frequency of lesions for age , and there are significant differences in the mean age between groups. Aging could be related to a higher incidence of lesions. Was there a relation in the frequency of type I and II lesions and age?
7 - The authors found significantly more caverns in CSC patients and hypothesize that choroidal thickening and hyperpermeability might explain this finding. Nevertheless, they do not correlate them to increased angiographic choroidal permeability areas or area of previous retinal serous detachment. They even assume that no angiographic findings were detected in the caverns’ areas (page 14, lines 289-290)
8 - The authors subdivide the choroidal caverns in 2 types depending on their size and location, but in discussion they do not elaborate any considerations on this. They do not speculate on their significance or hypothetical pathophysiological differences between them, if any. Could type I be precursors of type II lesions? To me this would be the point of differentiating the lesions.
9 - There are more significant limitations to be mentioned in this study as the manual segmentation of the lesions, the subjective bias of the observer identifying the lesions, the fact that only some patients performed an angiographic study, the lack of a confocal examination with the same device system for the OCT /OCTA and angiography.
10 - References 4 and 5 are completely inadequate and not referred along the manuscript:
“4 - Guo X, Wu Y, Wu Y, et al. Detection of superficial and buried optic disc drusen with swept-source optical 350 coherence tomography. BMC Ophthalmology. 2022; 22(1):219. DOI: 10.1186/s12886-022-02447-2. 351
5 - Guo X, Lei B, Gao Y. A Pigmented Vitreous Cyst Within the Posterior Precortical Vitreous Pocket. JAMA 352 Ophthalmology. 2022;140(1):e214681. DOI: 10.1001/jamaophthalmol.2021.4681. “
11 - Reference 8 is in germain
Round 2
Reviewer 1 Report
I'd like to thank the authors for the corrections made to the paper. I believe that the manuscript is greatly improved.